# Archaeometric analysis of Early Bronze Age bread from Küllüoba Höyük

**Salih Kavak** [1]*, **Yusuf Tuna** [2]©, **Yasin R. Eker** [3]©, **Şemsettin Akyol** [4]©,
**Abdurrahim C. Özcan** [5]©, **Murat Türkteki** [2]©

1 Department of Archaeology, Gaziantep University, Gaziantep, Turkey, 2 Department of Archaeology, Bilecik Şeyh Edebali University, Bilecik, Turkey, 3 Basic Science Department, Engineering Faculty, Necmettin Erbakan University, Konya, Turkey, 4 Archaeologist, Turkey, 5 Department of Prehistoric Archaeology, İstanbul University, İstanbul, Turkey

© These authors contributed equally to this work.
* salihkavak@gantep.edu.tr

## Abstract

Bread is a fundamental foodstuff that has driven social and technological development for millennia, with the earliest evidence dating to pre-agricultural societies. While archaeological sites from the Neolithic period show systematic grain processing, well-preserved bread from the subsequent Early Bronze Age, particularly in a clear ritual context, is exceedingly rare. Here we report the discovery and comprehensive archaeometric analysis; employing Scanning Electron Microscopy (SEM) coupled with Energy Dispersive X-ray (EDX) spectroscopy, Vibrational Spectroscopy (FTIR and Raman), and Thermal Analysis (TGA-DSC) of a 5,000-year-old carbonized bread from the Küllüoba settlement in Anatolia, dated 3200−3000 BC. Microscopic examinations reveal that it is made from coarsely ground emmer wheat (*Triticum dicoccum*) and a small amount of lentils (*Lens culinaris*). The presence of air voids suggests kneaded dough, possibly leavened. The detection of rachis fragments indicates the use of unsieved flour. Intentionally deposited and subsequently carbonized, the bread was sealed beneath a layer of sterile soil and appears to have been an offering connected with the ritual abandonment of the structure. This finding offers unique evidence of advanced food technology and highlights the symbolic importance of bread in Early Bronze Age societies, directly linking food production to cultural and ritual practices.

## Introduction

As one of the oldest and most fundamental foods in human history, bread is not merely a foodstuff but also a tool that strengthens social solidarity, identity, and societal organization [1]. Imbued with symbolic meanings throughout history, bread has played a central role in the relationship between communities, nature, and the sacred [2,3]. This study of the 5,000-year-old carbonized bread from Küllüoba highlights

**Data availability statement:** All relevant data are within the paper and its Supporting Information files.

**Funding:** The authors received no specific funding for the research, design, data collection, or analysis of this study. The Article Processing Charge (APC) will be covered by LESAFFRE TURQUIE MAYACILIK ÜRETİM ve TİC. A.Ş.

**Competing interests:** The authors have declared that no competing interests exist.

the symbolic importance of bread in Early Bronze Age societies, directly linking food production to cultural and ritual practices.

Hunter-gatherer societies possessed grain processing skills even before the advent of agriculture [4]. This process began with the grinding of plants in the Palaeolithic. With the domestication of wheat and barley, bread became a food product and a component of social identity, solidarity, and economic systems. Bread, therefore, became one of the symbols of sedentary life. Bread production processes have long been intertwined with ritual and social functions, as evidenced by 23,000-year-old grinding stones at the Ohalo II site in the Near East and ovens found in Neolithic settlements [4]. Pre-agricultural communities' complex grain processing techniques reveal bread's role beyond mere nutrition. Carbonized bread-like remains found at the Shubayqa 1 site in Jordan, dating back approximately 14,000 years, provide evidence of such advanced grain processing techniques [5,6].

With the Neolithic, bread production became systematized and integral to social life [7]. Ovens and grinding stones found in large-scale Neolithic centers, such as Çatalhöyük, show that bread-making contributed to developing the division of labor [8]. Associated with rituals, bread production played an important role in shaping social identity and cultural values [9].

However, the identification and characterization of amorphous charred food remains present significant challenges when relying solely on macroscopic observation. To address this, recent archaeobotanical research has increasingly integrated advanced archaeometric techniques to unravel the of ancient culinary practices. Methodologies such as Scanning Electron Microscopy (SEM) have become essential for visualizing microstructural details, including starch granules and aleurone layers, which are crucial for taxonomic identification and understanding grain processing. Complementing microscopy, chemical analyses using Vibrational Spectroscopy (FTIR and Raman) and Thermal Analysis (TGA-DSC) allow researchers to probe the molecular composition, detecting traces of proteins, lipids, and carbohydrates, and to estimate baking conditions such as temperature [5,10]. This multi-analytical approach represents the current state-of-the-art in food archaeology, enabling a precise reconstruction of ingredients and technological choices that visual inspection alone cannot provide.

This study examines the archaeological and archaeometric data from the carbonized bread remains discovered during the 2024 excavations at Küllüoba Höyük. The mound is located at the western end of the Upper Sakarya Basin, east of the province of Eskişehir. It is a medium-sized mound, measuring approximately 350x250 meters. The Küllüoba Höyük presents a continuous occupation sequence from the Transition to the Early Bronze Age through to the Transition to the Middle Bronze Age (ca. 3200–1900 BCE). This long-term settlement has been divided into six main layers based on stratigraphic radiocarbon results and relative chronology. The stratigraphic sequence begins with the lowest layers, corresponding to the advanced stages of the Transition to the Early Bronze Age (EBA Ia) and the subsequent Early Bronze Age I (EBA Ib). Following this, architectural remains attributed to the early (EBA IIa), middle (EBA IIb), and late (EBA IIc) phases of the Early Bronze Age II stand out, particularly

for their planned architectural layout, administrative units, storage and production areas, and indications of social organization (Table 1). The uppermost layers correspond to the Early Bronze Age III (EBA IIIa) and the Middle Bronze Age (EBA IIIb) transition. This stratigraphic sequence, established due to more than thirty years of excavations at Küllüoba, reveals in detail the processes of continuity and transformation in chronological and cultural terms. Moreover, it sheds light on early state formation and urbanization dynamics in Western and Inland Western Anatolia.

Recent excavations at Küllüoba have revealed structures deliberately filled with red soil [11] (Fig 1). This indicates that the custom of constructing ritual burials, primarily observed in Neolithic Anatolia, may have continued into the Early Bronze Age. The continuation of this practice, previously identified at Neolithic sites such as Göbekli Tepe, Karahantepe, and Çatalhöyük, as well as Chalcolithic sites like Canhasan [12–16], enhances our understanding of the social and ritual dynamics of the period.

Excavations in Area 2, located in front of Structure No. 1, have revealed a distinct architectural layout compared to other structures on the site. The building, consisting of two rooms, contained six ovens in Room 1, indicating that it underwent five separate construction phases. The bread, the subject of this study, was found on the lowest level of the structure. Four absolute radiocarbon dates from this phase (Layer VI) at Küllüoba indicate that this layer should be dated between 3200 and 2900 BCE (Fig 2). The C14 sample taken from the upper architectural phase of the room where the bread was discovered yielded the above result. The C14 analysis of the bread sample (see Fig 5) yields a date that significantly refines the stratigraphy.

**Table 1. The overview of the stratigraphy of Küllüoba Höyük.**

| Approximate Date Range (BC) | Stratigraphy |
|---|---|
| 2150−1950 | EBA IIIb |
| 2400−2150 | EBA IIIa |
| 2500−2400 | EBA IIc |
| 2600−2500 | EBA IIb |
| 2700−2600 | EBA IIa |
| 2900−2700 | EBA Ib |
| 3200−2900 | EBA Ia |

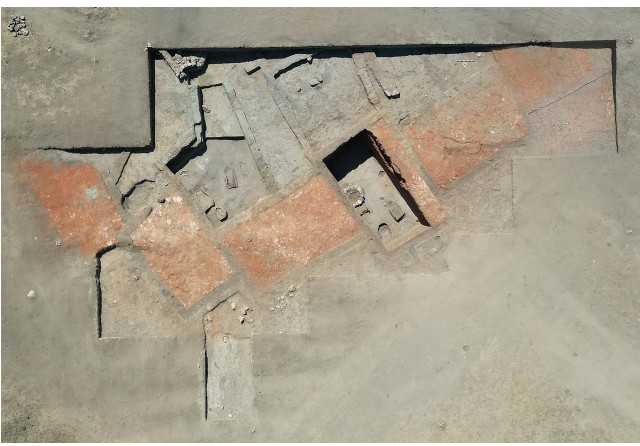

**Fig 1. An aerial view reveals the location of the buried structures.**

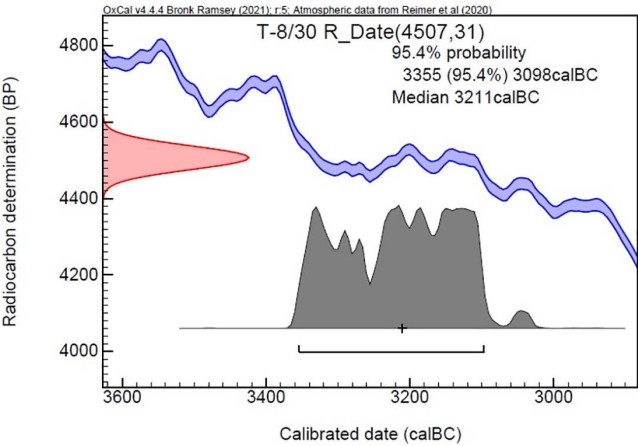

**Fig 2. Radiocarbon dating of Küllüoba Layer VI.**

## Materials and methods

The bread remains analyzed in this study is housed at the Eskişehir Eti Archaeology Museum in Eskişehir, Turkey. The specific inventory number for this specimen is 50/1. Analysis permits were obtained from the Republic of Türkiye Ministry of Culture and Tourism, General Directorate of Cultural Heritage and Museums. All necessary permits were obtained for the described study, which complied with all relevant regulations.

## Description of the Küllüoba bread

The most remarkable find in the room is the carbonized bread, discovered just east of the doorway (Fig 3).

The oval, bread measures 12 cm in diameter and 2.5 cm thick. It was recovered without damage during excavation. A piece was missing from its edge, suggesting it may have been deliberately broken off (Fig 4).

The location of the find and subsequent analysis indicate that the bread was deliberately carbonized as part of a ritual act and left as an offering, rather than being accidentally burned kitchen waste. Similar ritual practices are known from various periods [17–19]. The 50 cm thick layer of sterile red soil covering this phase protected the bread, allowing it to survive through subsequent occupation phases.

The bread remains were photographed and documented during the excavation work, then taken directly from the site to the research laboratory without undergoing any processing. Small sample pieces taken from the bread remains were examined and photographed using a stereozoom microscope in the laboratory. The sample taken for other spectrometric and compositional analyses was taken to the Necmettin Erbakan University Science and Technology Application and Research Centre, where the analyses were conducted in the laboratories there.

The surface observations were made using a Leica EZ4HD stereozoom microscope. The dough morphology and the microscopic fragments (granules, cells, phtoliths...) were characterized as is by high-resolution imaging obtained from field-emission scanning electron microscopy (Zeiss Gemini 500, FE-SEM). The accelerating voltage of electron was 2 kV while the FE-SEM images were obtained using secondary electron imaging (SE2 mode). The area elemental mapping of the sample was performed using an energy dispersive X-ray (EDX) detector (Oxford Instrument, Xmax 50 SEM) attached to the FE-SEM with an accelerating voltage of 7 kV using an SDD detector. The chemical composition was assessed by Fourier Transform Infra Red (FTIR) spectroscopy (Thermo Scientific Nicolet iS20). The spectrum was recorded in attenuated total reflectance (ATR) mode in the 4000−400 cm$^{-1}$ range with a resolution of 0,5 cm$^{-1}$.

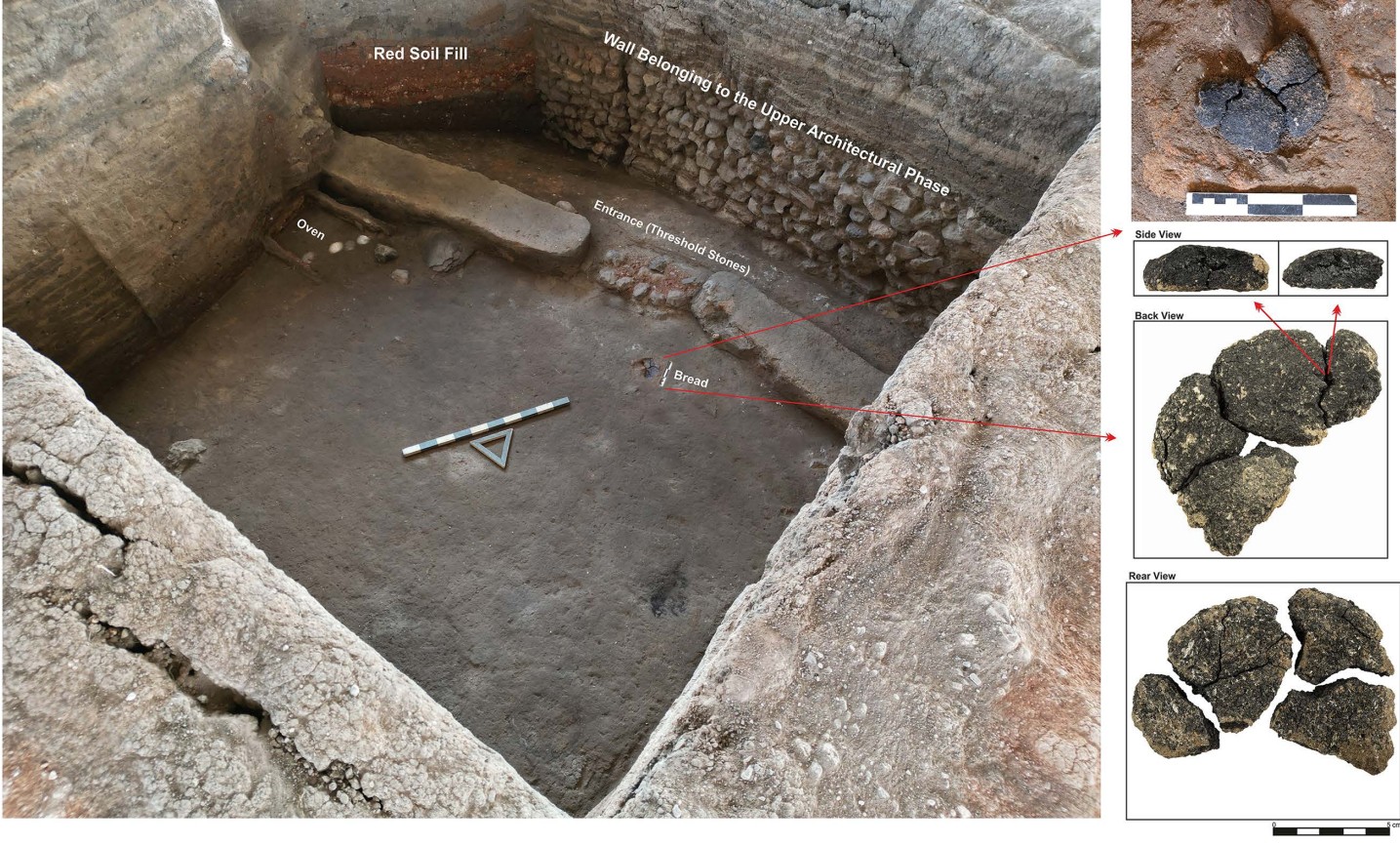

**Fig 3. Küllüoba Höyük excavation, Room 2, and the position of the bread unearthed.**

Only few micrograms of sample was analyzed without any specific preparation. Similar sample preparation was used to complete the chemical analysis via Raman spectroscopy (Renishaw inVia Reflex confocal Raman Microscope). The Raman spectrum was obtained with a 532 nm laser beam in the 800−3500 cm$^{-1}$ range with a resolution of 0,3 cm$^{-1}$. To improve the relevance of the results, both spectra were deconvoluted with Voigt functions using Fityk version 1.3.1 software. The detected peaks were finally compared and interpreted using the spectra of archaeological and contemporary breads investigated in the literature. Finally, the thermal behaviour of few micrograms of sample was performed with thermogravimetric analysis coupled with differential scanning calorimetry (TGA-DSC) instrument (Setaram Labsys Evo). The measurement was done from room temperature to 500°C at a heating rate of 10oC/min under nitrogen atmosphere (20 mL/min).

For archaeobotanical research, a total of 22 soil samples (62.85 L) were collected from Area 2, where bread was found. Carbonized plant remains were separated from the soil samples using a flotation system (motorless drum system). A soil sample is poured into the drum, and all light materials, including plant remains, rise to the surface. A constant flow of water passing through the system collects these floating materials in a chiffon located at the part of the drum where the water exits. These samples were then dried, sifted, and packaged for analysis. The samples were examined, identified, and classified under a stereozoom microscope at the Istanbul University Prehistory Laboratory.

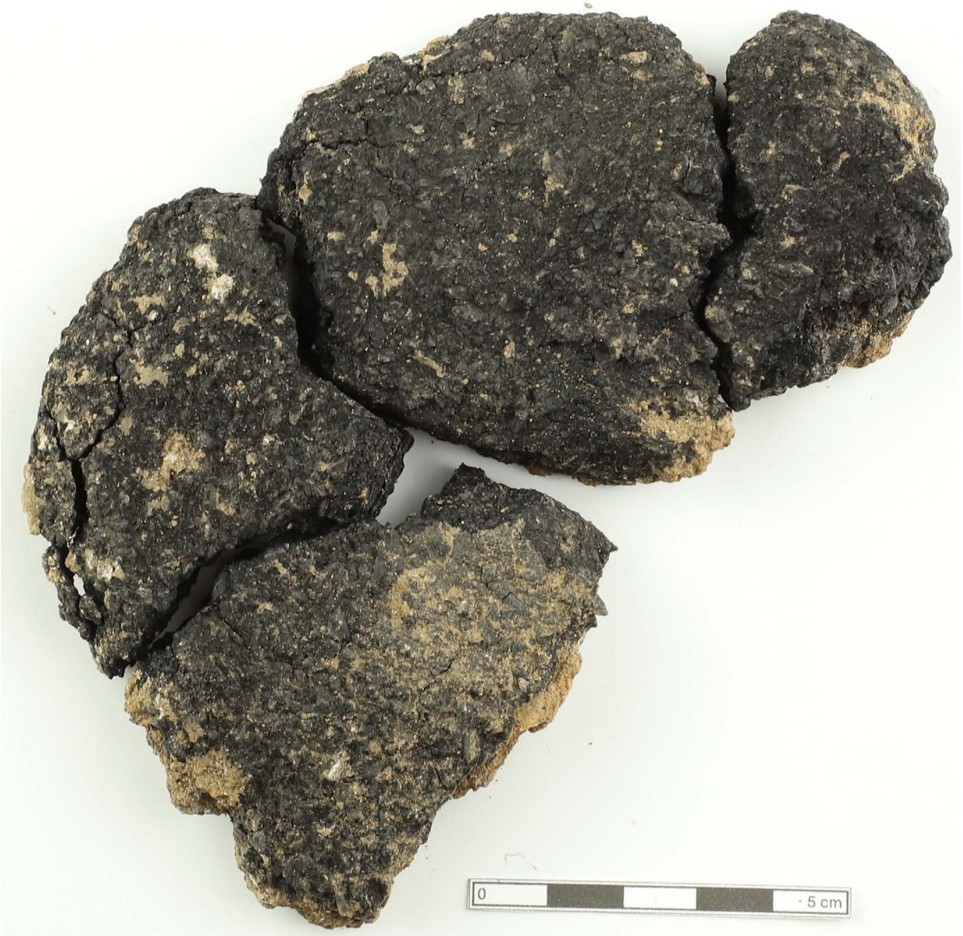

**Fig 4. Küllüoba bread remains.**

## Results and discussion

### Radiocarbon (C¹⁴) dating

The radiocarbon dating of the bread was performed at the TÜBİTAK Marmara Research Center Accelerator Mass Spectrometry (AMS) laboratory. The standard radiocarbon age for the sample obtained from the bread remnant (T8-50/2) was established as 4385+/- 34 BP. The calibrated outcomes at a 95.4% probability reveal intervals of 3261–3252 cal BC (0.8%) and 3100–2907 cal BC (94.7%). The median date was determined to be 2993 cal BC. The data definitively situate the material in the late 4th millennium BCE (Fig 5).

### Microscopic analysis

SEM analyses revealed air voids of various sizes and irregular shapes on the surface and inside the bread (Fig 6). These voids indicate that the dough was kneaded and subjected to brief heat treatment.

Microscopy revealed coarsely ground grain particles distributed throughout the inner matrix, not just on the surface (Fig 7).

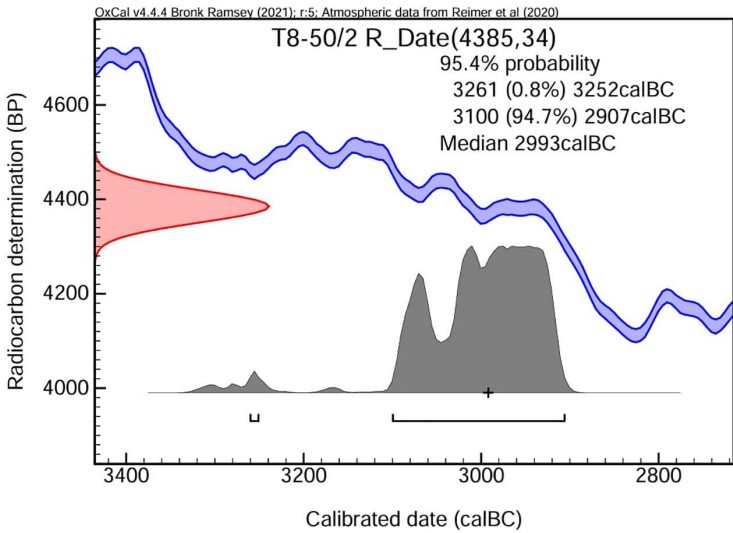

**Fig 5. Radiocarbon dating of Küllüoba bread.**

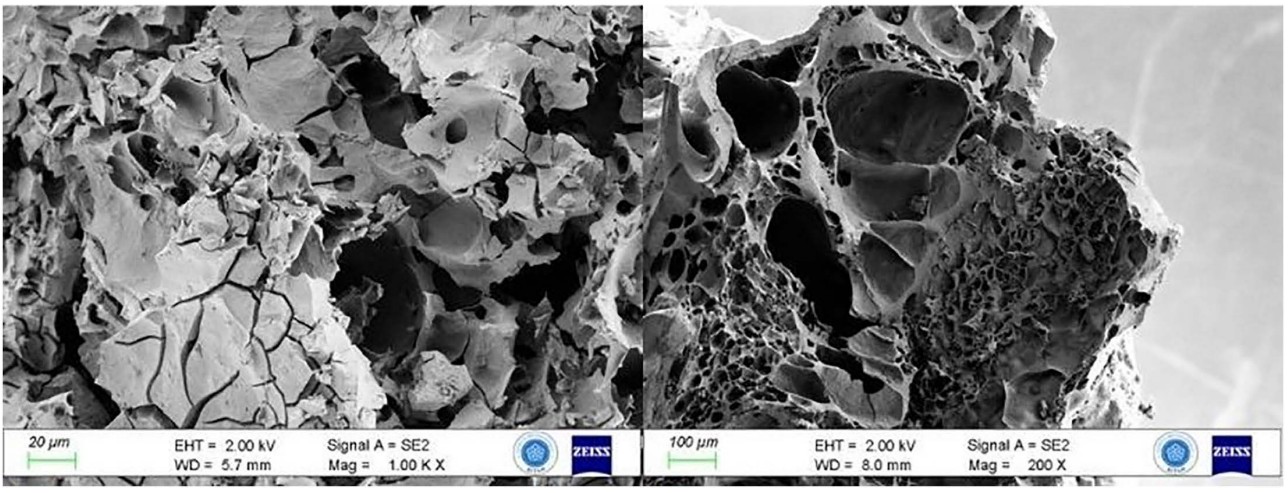

**Fig 6. SEM images of bread dough texture.**

Microscopic examination revealed seed fragments and bran particles exhibiting morphological characteristics indicative of *Triticum sp.* (Fig 8). Upon evaluating the micromorphological observations alongside the archaeobotanical data from the site, where *Triticum dicoccum* (emmer wheat) is the principal cereal crop found in the hearths and surrounding environment, emmer wheat is recognized as the primary component of the bread. This demonstrates that the main raw material was emmer wheat, which was widely grown in early agricultural societies.

In particular, the bread contained not only cereal but also legume residues. Stereomicroscope data and SEM images indicated structures matching *Lens culinaris* (lentil) based on the morphology of ground seed fragments and seed coats (Fig 9). This suggests the bread's recipe included legumes, albeit in small amounts, to increase its nutritional value.

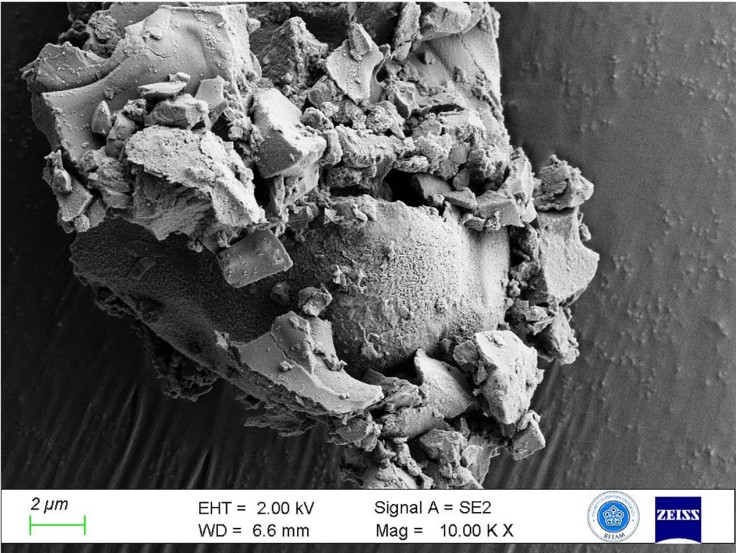

**Fig 7. Cereal grain ground into a dough-like consistency.**

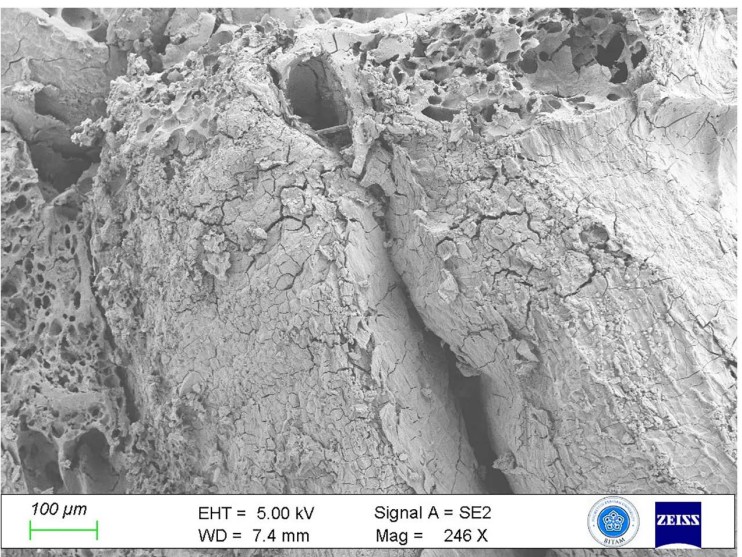

**Fig 8. Grain fragment of *Triticum dicoccum* (emmer wheat).**

A rachis (spikelet stalk) fragment is also found in the bread (Fig 10). This finding reveals that the grains were used in a relatively unprocessed state, without being fully sieved, indicating a functional yet straightforward processing technology.

Detailed SEM analysis showed that the starch granules had different shapes and sizes that matched the cereal and legume species that were found (Fig 11). The preservation of these granules shows that cereal and legume carbohydrates retained their structural integrity after cooking. This suggests the bread may have been baked briefly at a high temperature.

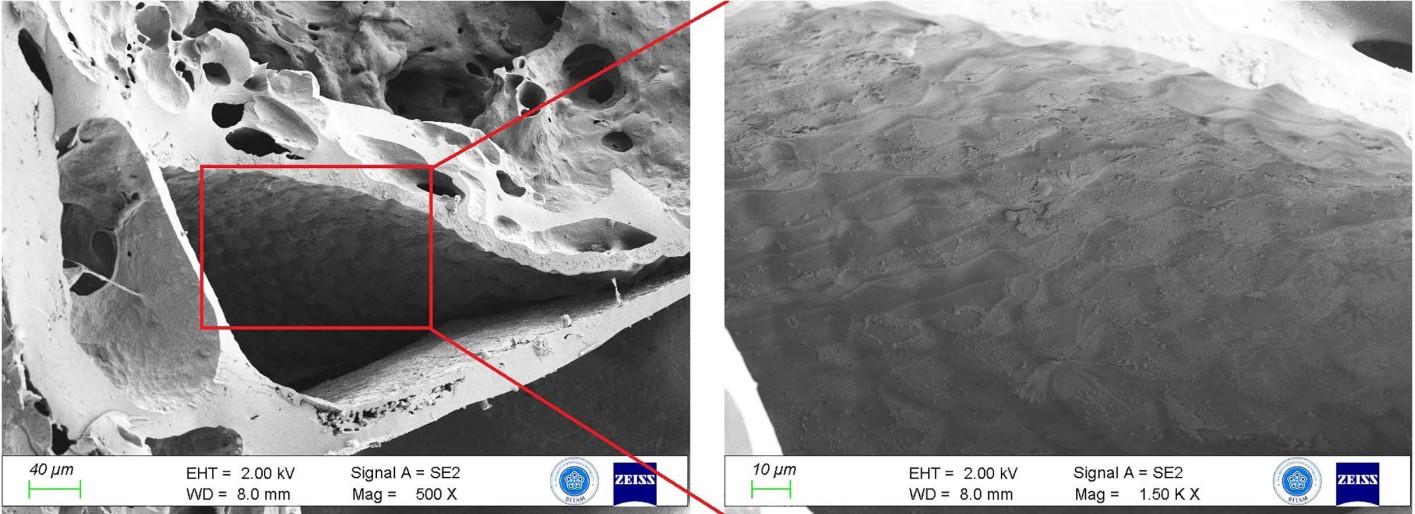

**Fig 9. SEM image of the seed coat of Lens culinaris (lentil).**

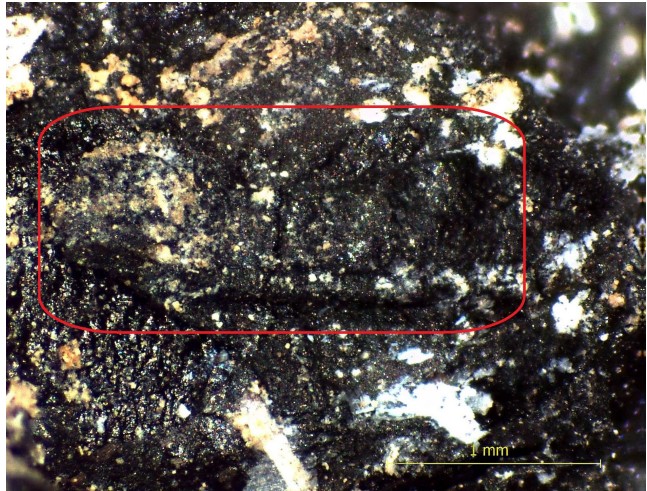

**Fig 10. Rachis fragment in bread tissue under a stereozoom microscope.**

In addition to analyzing the microscopic structure, the sample's chemical composition was analyzed using EDX (Fig 12). The analysis identified carbon (C) as the dominant element, with concentrations ranging from 49.4% to 59.8%, which is consistent with the carbonized nature of the organic material. Oxygen (O) was the second most abundant element, detected at 25.3% to 29.2%. The prevalence of these elements confirms the organic origin of the sample and is typical for carbonized plant-based materials. Calcium (Ca) was also detected at lower levels (6.2% to 10.9%). This element likely resulted from contamination from the surrounding soil, as minerals like calcium carbonate ($CaCO_3$) may have adhered to the bread's surface.

These multifaceted analyses demonstrate that the Küllüoba bread is not merely a carbonized residue but also preserves microstructural and elemental traces of its plant components. Combining *Triticum dicoccum* and *Lens culinaris*

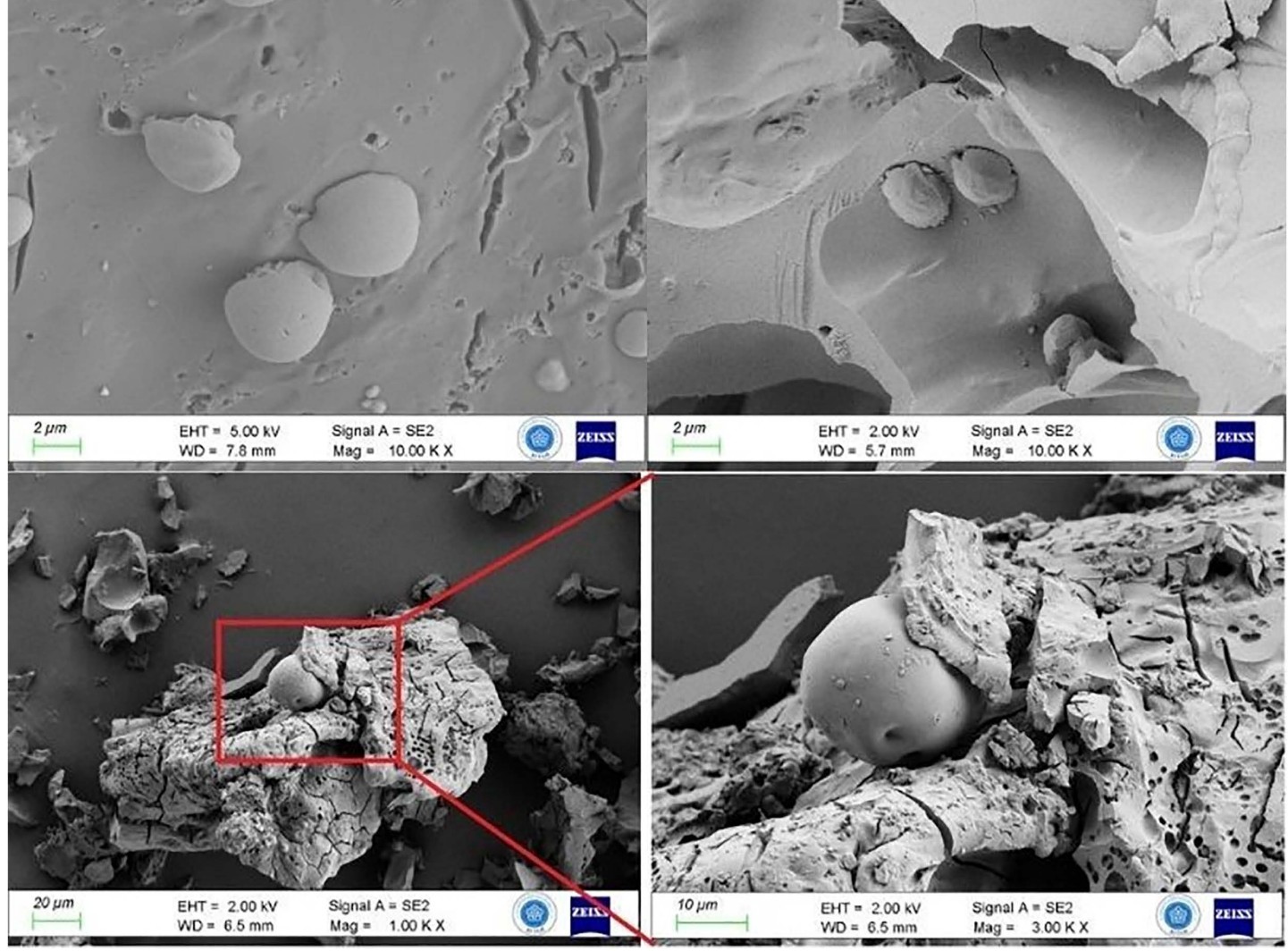

**Fig 11. Different morphologies and sizes of starch granules.**

suggests a nutrient-dense and deliberately diversified recipe. The presence of rachis fragments and varied particle sizes suggests that unsieved, coarsely ground raw materials were used, resulting in a bread product that is both technologically simple and nutritionally rich.

## Vibrational spectroscopy

Vibrational spectroscopy is a chemical characterization technique based on irradiation and time-limited excitation of the sample. Fourier Transform InfraRed (FTIR) spectrum is obtained via infrared (IR) light radiation, and Raman spectrum is obtained with monochromatic light radiation.

The chemical structure of the bread sample was investigated by both FTIR and Raman spectroscopy. Since the chemical structure of the initial dough ingredients is unknown and the sample was stored underground for a long time, all peaks are probably not specific to the bread sample. Therefore, the interpretation of the peaks can be discussed again with the

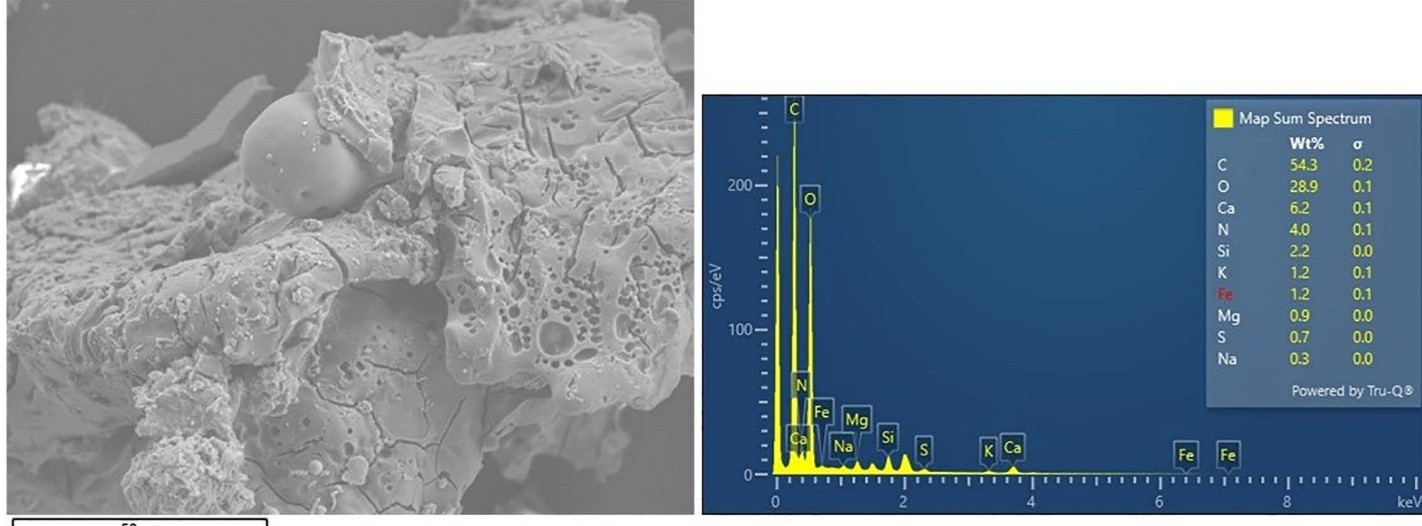

**Fig 12. Result of chemical analysis of dough tissue using EDX.**

development of further information. To improve the relevance of the results, both spectra were deconvoluted with Voigt functions using Fityk version 1.3.1 software. The detected peaks were finally compared and interpreted using the spectra of breads investigated in the literature.

The FTIR spectrum presents three significant peak domains between approximately 800–1200 cm$^{-1}$, 1300–1800 cm$^{-1}$, and 2800–3700 cm$^{-1}$, and two other peak domains between 400–800 cm$^{-1}$ and 1800–2400 cm$^{-1}$ (Fig. 13). Peaks below 800 cm$^{-1}$ are related to the skeletal vibrational modes of aromatic rings, while those between 1800–2400 cm$^{-1}$ represent conjugated double or triple bonds between C, N, and/or O [20,21]. Both domain peaks confirm that a large number of organic molecules with bond resonances in their structure are present in the sample.

Among the most important peak domains, a large number of peaks between 800–1200 cm$^{-1}$ (1135, 1098, 1039, 1007, 1005, 990, and 921 cm$^{-1}$) are characteristic for polysaccharide molecules (Fig 14a). The most intense peaks at 1005 and 1007 cm$^{-1}$ correspond to those of starch [21].

In the 1300–1800 cm$^{-1}$ region, peaks at 1627, 1551, and 1422 cm$^{-1}$ are typical of the stretching bands of amide I, II, and III found in gluten and proteins (Fig 14b) [22,23]. Furthermore, the peak at 1657 cm$^{-1}$ is probably related to the -OH bending of adsorbed water and not to the bread structure. Otherwise, no peaks were detected between 1700–1800 cm$^{-1}$, which usually reflects the presence of carbonyl groups. The absence of carbonyl groups indicates that the organic structure has been heavily carbonized.

In this case, the absence of carbonyl groups can be attributed to either the lack of oil in the dough, which leads to the loss of carbonyl groups at high temperatures [24]. Or it can be explained by the low water content in the dough recipe, since its excess promotes ester hydrolysis, which favors the presence of carbonyls in the structure [22].

Finally, the peaks in the 2800–3700 cm$^{-1}$ region are related to single bond (N-H, O-H, C-H…) stretching modes (Fig 14c). The peak at 3407 cm$^{-1}$ is characteristic for OH in carbohydrates, proteins, and polyphenols [20]. The peak at 3301 cm$^{-1}$, the lowest wavenumber, reflects the intermolecular H-bonded O–H stretching, while the peaks at 3539 and 3618 cm$^{-1}$, the highest wavenumbers, are due to the asymmetric O-H stretching specific to non-hydrogen-bonded isolated hydroxyl. Isolated OH is present in the protein structure of the gluten network when bread is well baked [22]. Finally, peaks at 2927 and 2862 cm$^{-1}$, typical of the C-H stretching mode, are observed in breads, and their attenuation reflects the presence of oil and/or salt, which are critical for the development of the gluten network [24].

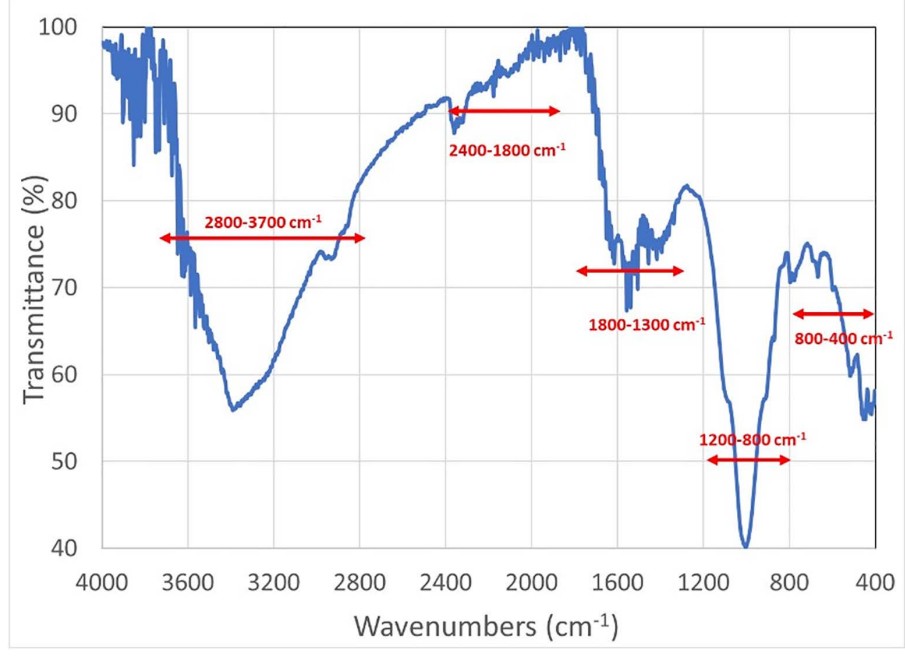

**Fig 13. Experimental FTIR spectrum of the Küllüoba bread sample.**

Comparison of the FTIR spectrum with those of bread literature supports the presence of lipids, polysaccharides, and proteins. In particular, peaks at 2927 and 3301 cm$^{-1}$ may be typical of the gluten network (2921 and 3280 cm$^{-1}$ according to Sivam et al. [22]). On the other hand, the absence of carbonyl groups and the attenuated C-H stretching modes suggest that the bread dough was prepared with a minimum amount of water.

The Raman spectrum presents significant peak regions deconvoluted by the combination of strong and weak peaks (Fig 15). Among the strongest peaks, those at 1141, 1272, 1373, 1549, 1601, and 2914 cm$^{-1}$ are due to specific bonds, while the one at 2642 cm$^{-1}$ is an overtone reflecting the vibrational behavior of single aromatic layers, as in the single-layer graphene [25,26]. Therefore, the presence of carbohydrates (1141 and 1272 cm$^{-1}$ C-O stretching of ether), proteins (1373 cm$^{-1}$ N-H bending amide III, 1549 cm$^{-1}$ N-H bending amide II, and 1601 cm$^{-1}$ C=C stretching aromatic), and lipids (1601 cm$^{-1}$ C=C stretching aromatic and 2914 cm$^{-1}$ =C-H stretching in saturated lipids) was predicted [22,27]. On the other hand, the weakest peaks at 1358 cm$^{-1}$ (CH$_2$ bending in Aliphatic), 1740 cm$^{-1}$ (C=O stretch of ester/carboxylic acid in lipids and amino acid), 2438 cm$^{-1}$ (-C≡N stretch of Nitrile), and 3214 cm$^{-1}$ (-C-H stretch in unsaturated lipids) indicate that other organic structures are present in smaller amounts. Additives are probably enriching the nutritional quality of the breads.

### Thermal characteristics

On the thermogram up to 500°C, half of the total mass loss (~13%) occurs below 150°C. This temperature value is critical for bread, where the expected mass loss is approximately 28%, while for dough this value is 36% [28,29] (Fig 16). The difference in mass loss between ancient and recent bread can be explained by the slow carbonization process. However, 10% of the mass loss was observed around 100°C due to the soil moistening over time. This aspect is supported by endothermic heat flow, indicating the breaking of weak bonds and the evaporation of moisture [30]. Above 150°C, following a significant endothermic peak, a large exothermic peak seems to begin between 160°C to 230°C. Furthermore, mass loss

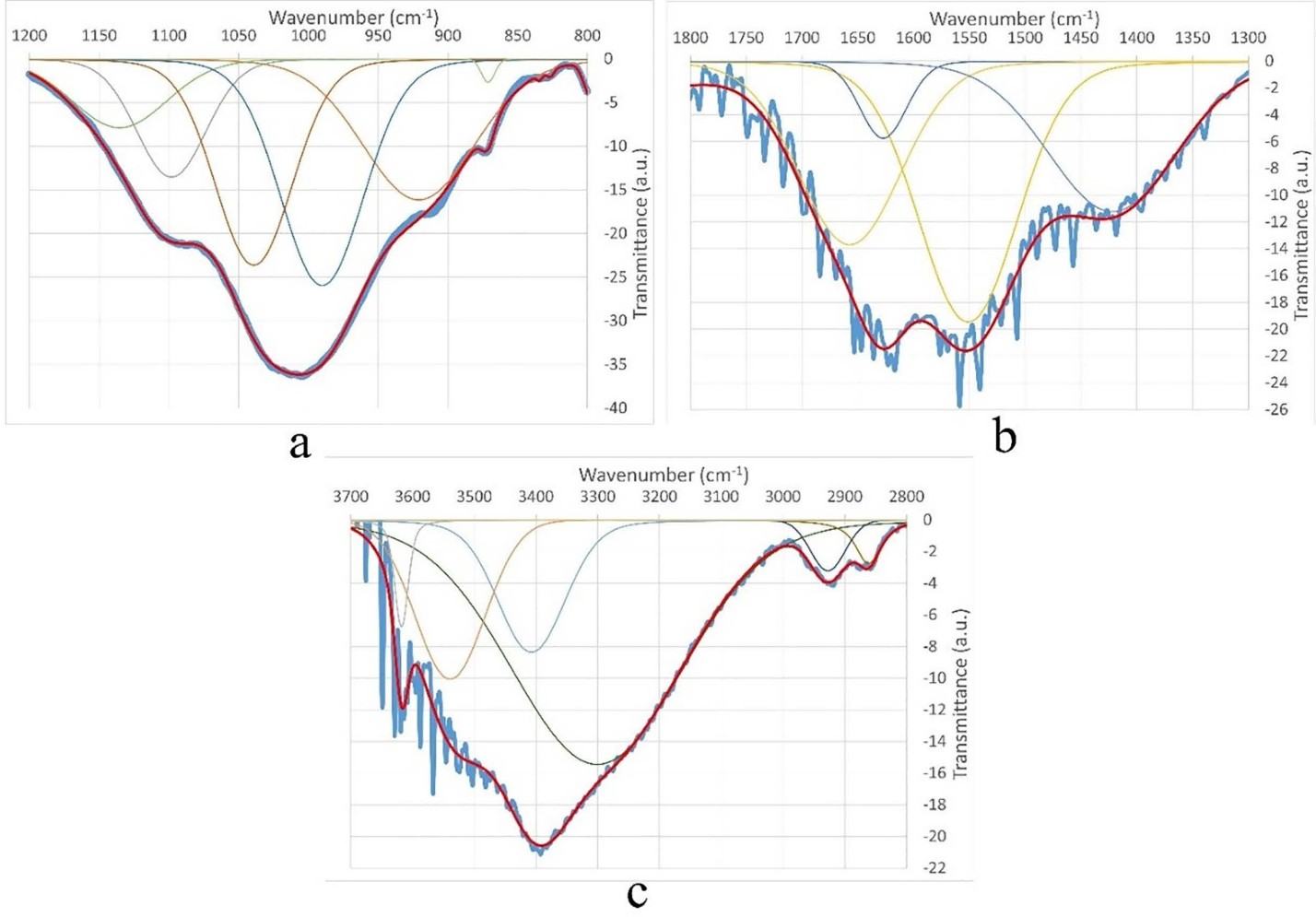

**Fig 14. Deconvoluted FTIR spectrum domains of the Küllüoba bread sample (a) 800-1200 cm$^{-1}$ (b) 1300-1800 cm$^{-1}$ (c) 2800-3700 cm$^{-1}$ (Experimental spectrum in blue – Simulated spectrum in red).**

continues during heating and a new inflection can be discerned between 160 and 250ºC. These results are likely related to the formation of new bonds and the release of small molecules, indicating that the sample had not previously been heated to this temperature.

## Archaeobotanical Results

The archaeobotanical samples examined were taken from inside hearths, in front of hearths, inside pots/bowls, next to ceramics, from pits, and various areas/corners within the structure.

## Hearths

Hearths are one of the most obvious archaeological sites indicating cooking activities and the use of plant remains within structures. Four archaeobotanical samples (T8: 32/1, 44/1, 45/2, and 48/1) were selected and examined from this site (S1 Table). According to the data obtained, grain remains were found in high concentrations in the hearths. These grains

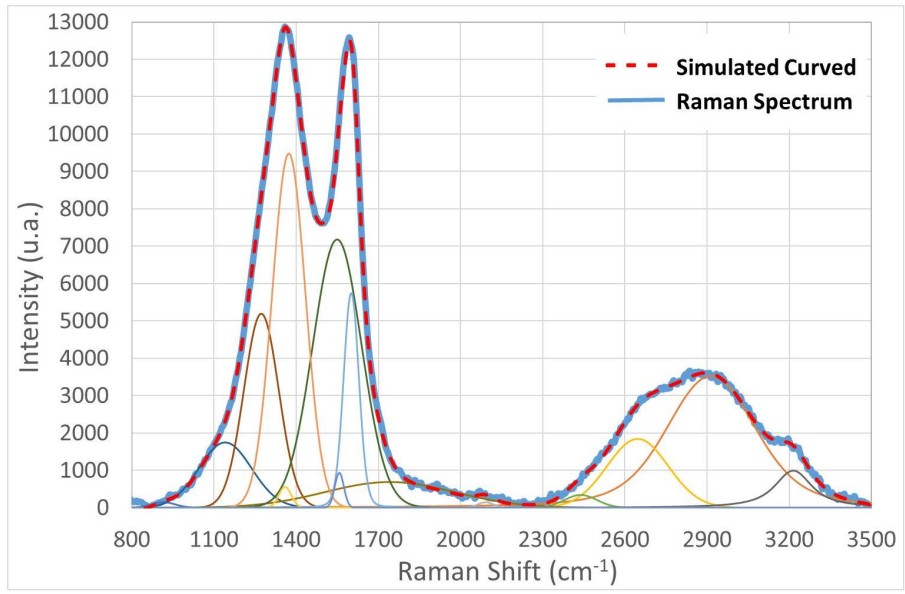

**Fig 15. Deconvoluted Raman spectrum of the Küllüoba bread sample.**

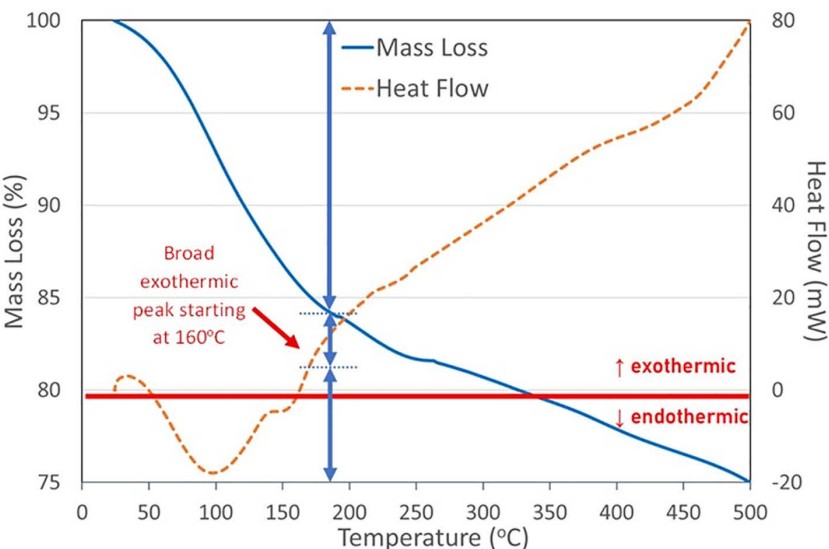

**Fig 16. TGA-DSC curve of Küllüoba breads.**

include emmer, einkorn, six-row barley, and bread/durum wheat (Fig 17). Grains in the hearths indicate that these products were included in the cooking process and that an attempt was made to prepare bread or grain-based foods. But they may have been charred by careless use, and the leftover grains may have been mixed with fuel in the hearths.

Additionally, when examining the wheat and barley obtained from the hearths above, it is worth noting that they exhibit severe burns that could cause significant morphological abnormalities. In addition to consumption, the extraction of einkorn/emmer and six-row barley rachis from the stoves indicates that the grains underwent specific processes (such

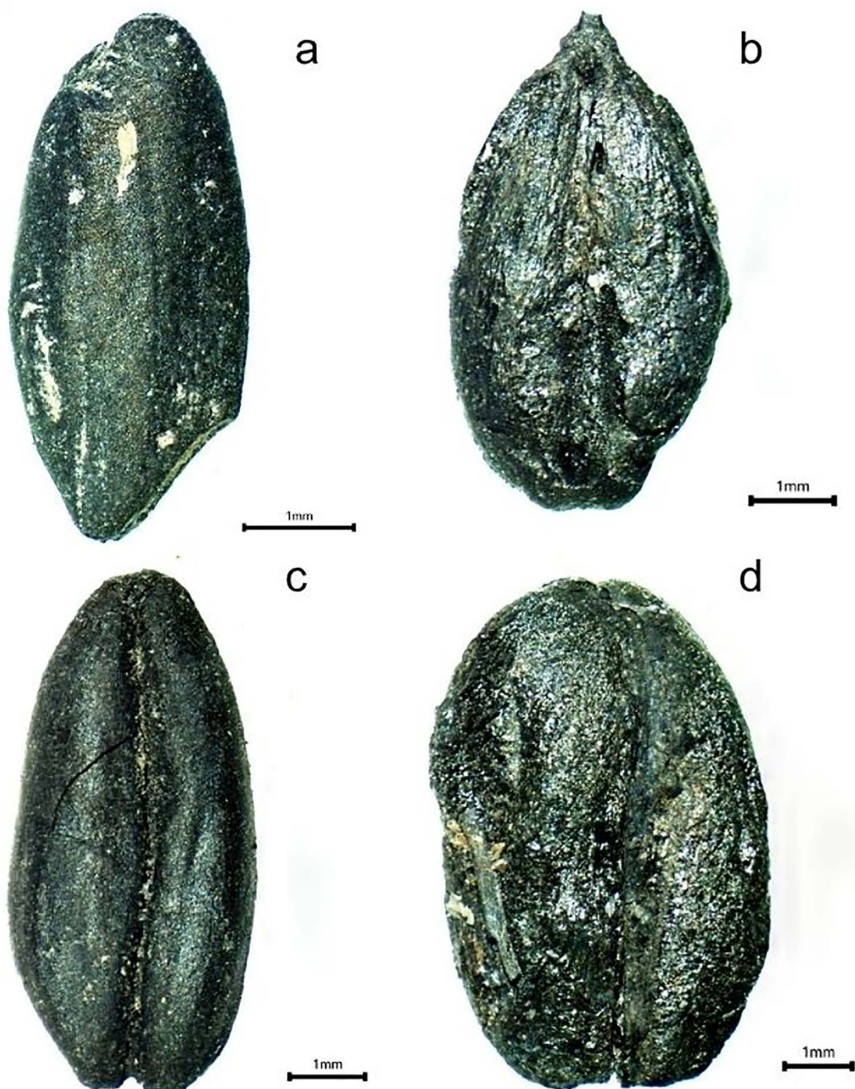

**Fig 17. Cereals of Area 2.** (a) einkorn (*Triticum monococcum*), (b) barley (*Hordeum vulgare*), (c) emmer (*Triticum dicoccum*), (d) bread/durum wheat (*Triticum aestivum*/*durum*).

as separation from their husks) within Area 2, and we can say that the rachis were considered waste and used for fuel purposes. In the examples, legume residues such as bittervetch, peas, and lentils are much less common than grains (Fig 18). The aforementioned legume residues may have been included in the consumption chain for food types such as soups, porridge, or bread within Area 2. Additionally, wild plants inside the hearths were mixed with the products to be burned (consciously or unconsciously). This is because many of these wild plants are considered weeds in fields.

Looking at the samples taken from the hearths, we can see that they have undergone cleaning processes. For example, the presence of botanical remains in sample 32/1, albeit somewhat incomplete, indicates that these areas underwent cleaning procedures during a specific period. However, when we compare this sample with another (T8: 48/1), the furnace was used without a complete cleaning procedure.

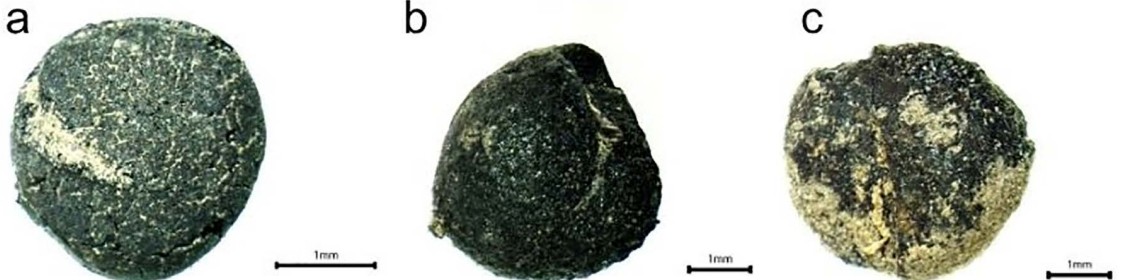

**Fig 18. Legumes of Area 2.** (a) lentil (*Lens culinaris*), (b) bittervetch (*Vicia ervilia*), (c) pea (*Pisum sativum*).

### In front of the hearths

To better understand the activities inside the hearths, samples directly related to the areas in question were taken from in front of the hearth areas and analyzed (T8: 38/1 and 45/3). These samples were generally obtained with an ashy structure. These archaeobotanical samples were collected in front of the hearths; their contents (such as types of grains and legumes) are very comparable to one another and do not differ much from those collected inside the hearths. These plant remnants that were removed from the hearth fronts must have been dispersed around the region as rubbish while the hearths were being cleaned.

### Pits

The pits inside the structures are where waste was accumulated and later disposed of, so the data inside them is always carefully observed. Archaeobotanical samples (T8: 37/1, 37/2, 39/1, and 48/1) from the four pits in Area 2 were analyzed to evaluate plant-based consumption within the space from different angles. The analyses revealed the presence of emmer, einkorn, and bread/durum wheat belonging to the cereal family and six-row barley residues. Legumes such as peas, bitter vetches, and lentils were also found in much smaller proportions than cereals. The abundant presence of rachis residues in the waste pits indicates that rachis was swept or thrown into the pits as waste/trash.

### Pottery Interiors and Surroundings

Since a systematic silo and storage area concept has not yet been identified in Area 2, archaeological botanical samples (T8: 32/5, 35/2, 36/11, 36/12) were taken from the ceramics and evaluated to understand storage strategies and identify plant use. Samples 32/5 and 36/11 were taken directly from the pottery, but only a small amount of botanical remains were found. However, a larger quantity of botanical remains was found in the sample collected from the area adjacent to pottery 34/4. The plant remains may have been mixed into the soil due to spreading from the pottery (perhaps through breakage).

In archaeological settlement contexts, particularly those reflecting daily life, food-related activities such as preparation and cooking are known to have occurred. We note that we have found similar evidence in Area 2. One of our findings was observed in an archaeobotanical sample taken from a pot in front of the fireplace, numbered 35/2. This food residue, which does not appear to be porridge or bread-like, has been subjected to a cooking process and is likely to contain black cumin residues. It has also been observed that the black cumin seeds in the food residue do not have seed coats. Even a small amount of black cumin residue in the archaeobotanical sample supports this idea. Another finding indicating that the food residues had been cooked was detected in an archaeobotanical sample (44/1) taken from the hearth. The thin food pieces resemble the flatbread commonly consumed today in terms of structure. However, there is also the possibility that they are crumbs. Observing whether the broken pieces contained grains or legumes was impossible. The food fragments

were exposed to high heat, causing their inner parts to become highly carbonized. Another porridge-like food residue, sample 46/3, was obtained from a botanical sample taken from the eastern side of the hearth. This food sample may also have been scattered from a handled vessel on the east hearth.

Another example of food remains comes from a garbage pit. In archaeobotanical samples taken from two different levels (top and bottom) of the same pit, no food remains were found at the bottom, but some organic remains were found at the top. The food samples in question consist of compacted grain fragments, and these organic remains were likely subjected to food preparation processes. In another sample from a different waste pit (sample no. 39/1), there are clumps of fragments consisting mainly of grains. The presence of hulled six-row barley residues, coupled with the observed alteration of morphological characteristics in both barley and certain durum wheat due to high heat exposure, indicates that these clump samples likely coalesced in a location such as a hearth and were subsequently discarded into the pit as refuse.

When looking at plant consumption in general, six-row barley, emmer wheat, einkorn wheat, and bread wheat are used in noticeable amounts in Area 2. Six-row barley is used with and without husks, while einkorn wheat is consumed as single-grain and double-grain varieties. The economic plant residues obtained directly support particular grain and legume residues in the bread's composition. Legume residues (lentils, peas, and bitter vetches) appear in smaller quantities than grains. This fact may be because legumes are stored in smaller amounts and for shorter periods than grains. This possibility is because legumes' water/moisture content makes them more prone to dampness, insect infestation, and decay. Another possibility is that legumes are used in liquid dishes without being ground. Bedstraw (*Galium sp.*), field gromwell (*Lithospermum arvense*), bromegrass (*Bromus sp.*), catchfly (*Silene sp.*), needle spikerush (*Eleocharis sp.*), and saltwort (*Salsola sp.)* are some of the wild plant residues observed in this category. Plants such as milkwort and lithospermum are known as field weeds and generally mix into the harvest during grain/legume harvesting. Therefore, their presence in the structure may be possible. The abundance of some examples of bromegrass (e.g., pit no. 39/1) raises questions about whether this wild grain species was also included in consumption. An example of plant remains collected from the environment is bramble (*Rubus sp.*). The presence of brambles in Area 2 indicates that the Küllüoba people did not consume only cereals/legumes but also collected brambles in season and ate their fruit along with the seeds, benefiting from the ecological system around them. Due to their fragile structure, detecting bramble fruits is usually a matter of chance [31].

The analysis results we obtained are primarily similar to the earlier archaeobotanical analyses of the Early Bronze Age I in Küllüoba [32]. The consumption of cereal and legume families and the similarity of wild plants have drawn a parallel pattern [33,34].

Microscopic and SEM analyses revealed that the main components of the Küllüoba bread were *Triticum dicoccum* (emmer wheat) and a small amount of *Lens culinaris* (lentil). Archaeobotanical studies from the same site support this finding. The abundance of barley in the site's archaeobotanical samples shows that it was not used in bread-making, indicating a preference for emmer wheat in bread production. This data offers substantial information about local preferences compared to studies from sites like Neolithic Çatalhöyük, which show different cereal mixtures or a focus on specific cereals [35,36]. Bread and food remains from Çatalhöyük indicate various grains and legumes, but specific recipes could vary temporally or spatially [35,37]. For comparison, 14,400-year-old bread-like remains from the Natufian settlement of Shubayqa 1 in Jordan indicate the use of wild cereals, such as barley, einkorn, and oats, as well as aquatic plant tubers [5]. This suggests that even pre-agricultural communities developed complex recipes [5], whereas the more refined recipe at Küllüoba, based on domesticated species, reflects the evolution of settled agricultural societies.

The variable-sized air pockets in the bread suggest that the dough may have undergone fermentation or leavening. This could indicate a significant technological step, especially compared to the unleavened flatbread from Shubayqa 1 [5] and the porridges and simple baked doughs found at many other archaeological sites [35,37,38]. Bran and coarsely ground particles suggest that whole grains were ground and used without sieving, resulting in a highly fibrous dough.

Thermal analysis indicated cooking temperatures above 150–160°C, providing indirect information about the oven or hearth technologies of the period.

The Küllüoba bread is described as oval and flat, with a diameter of 12 cm. This shape generally resembles the flatbreads from Shubayqa 1 [5], but differs in size and shape from the bread found at Çatalhöyük [37]. The size and form of the Küllüoba bread is unique among the known examples from Anatolia. Perhaps the most striking aspect of the discovery is the context. The bread's discovery near a doorway, deliberately carbonized and sealed under a sterile red earth fill, suggests it was more than a remnant of a daily meal. The ritualistic burial of structures is a practice known from the Neolithic and Chalcolithic periods [13–16], and the location strengthens the interpretation that the Küllüoba bread was part of an abandonment or closure ritual, perhaps as an offering. This aligns with the general understanding that bread carries nutritional but also social and ritual meanings [1–3]. While bread from other sites is often found in more domestic contexts, such as hearths or storage areas [5,35,37,39], this particular context at Küllüoba provides rare evidence of the symbolic value of bread in the Early Bronze Age.

## Conclusions

The carbonized bread remains from Küllüoba Höyük, dating to the Early Bronze Age (3200−3000 BC), is a significant find that sheds light on the history of bread-making and related cultural practices, not only in Anatolia but across the Near East. The archaeometric analysis of this bread reveals unique aspects of Küllüoba's food culture, allowing for comparison with finds from contemporary and earlier settlements. The microscopic and spectroscopic data have enabled detailed inferences about the production process, the raw materials, and baking techniques, providing crucial evidence for evaluating the find's ritual context.

The Küllüoba bread, with its specific ingredients, possible leavening, distinct form, and especially its ritualistic context, makes it an excellent point of comparison for the history of bread in the Near East and Anatolia. From the pre-agricultural Shubayqa 1 [5] to Neolithic Çatalhöyük [35,37] and Jarmo [38] and sites in Europe [39], the Küllüoba archaeological record illuminates the use of domesticated grains, basic processing techniques, specific grain preferences, and a complex ritual purpose. This detailed data, obtained through multiple analytical methods, confirms that bread is not only a food source but also an indicator of technological development and complex social behavior. The Küllüoba bread is one of the earliest known examples in Anatolia, both in terms of form and content, and it constitutes a unique record of the food culture of early societies.

## Supporting information

**S1 Table. Archaeobotanical data from Küllüoba Höyük Trench T8.**
(DOCX)

## Acknowledgments

The Küllüoba Excavation was supported by Bilecik Şeyh Edebali University Scientific Research Projects GAP-2024–568 No. Project. We would like to thank Dr. Ashley Cercone for reviewing and proofreading the article.

## Author contributions

**Conceptualization:** salih kavak, Yasin R. Eker, Murat Türkteki.

**Data curation:** salih kavak, Yusuf Tuna, Yasin R. Eker, Abdurrahim C. Özcan, Murat Türkteki.

**Formal analysis:** salih kavak, Yasin R. Eker, Murat Türkteki.

**Investigation:** salih kavak, Yusuf Tuna, Yasin R. Eker, Şemsettin Akyol, Abdurrahim C. Özcan, Murat Türkteki.

**Methodology:** salih kavak, Yasin R. Eker, Abdurrahim C. Özcan, Murat Türkteki.

**Project administration:** Murat Türkteki.

**Resources:** salih kavak, Murat Türkteki.

**Software:** Yasin R. Eker.

**Supervision:** salih kavak.

**Validation:** salih kavak.

**Visualization:** salih kavak, Yusuf Tuna, Yasin R. Eker.

**Writing – original draft:** salih kavak, Yusuf Tuna, Yasin R. Eker, Abdurrahim C. Özcan, Murat Türkteki.

**Writing – review & editing:** salih kavak, Murat Türkteki.

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
