## [Decision Letter · Decision Letter 0]

24 Nov 2025

Dear Dr. kavak,

Thank you for submitting your manuscript to PLOS ONE. After careful consideration, we feel that it has merit but does not fully meet PLOS ONE’s publication criteria as it currently stands. Therefore, we invite you to submit a revised version of the manuscript that addresses the points raised during the review process.

**All comments need to be addressed in detail before re-submission.**

We look forward to receiving your revised manuscript.

Kind regards,

Peter F. Biehl, PhD

Academic Editor

PLOS ONE

Journal Requirements:

2. In your manuscript, please provide additional information regarding the specimens used in your study. Ensure that you have reported human remain specimen numbers and complete repository information, including museum name and geographic location.

For more information on PLOS One's requirements for paleontology and archeology research, see https://journals.plos.org/plosone/s/submission-guidelines#loc-paleontology-and-archaeology-research ..

4. Please upload a copy of Figure 17, to which you refer in your text on page 14. If the figure is no longer to be included as part of the submission, please remove all reference to it within the text.

5. Please include captions for your Supporting Information files at the end of your manuscript, and update any in-text citations to match accordingly. Please see our Supporting Information guidelines for more information: http://journals.plos.org/plosone/s/supporting-information ..

Additional Editor Comments:

Your manuscript has now been seen by a referee, whose comments are appended below. You will see from these comments that while the referees find your work of potential interest, they have raised substantial concerns that must be addressed. In light of these comments, we cannot accept the manuscript for publication, but would be interested in considering a revised version that addresses these serious concerns.

We hope you will find the referees' comments useful as you decide how to proceed. Should presentation of further data and analysis allow you to address these criticisms, we would be happy to look at a substantially revised manuscript. However, please bear in mind that we will be reluctant to approach the referees again in the absence of major revisions.

Reviewer's Responses to Questions

**Comments to the Author**

1. Is the manuscript technically sound, and do the data support the conclusions?

Reviewer #1: Partly

2. Has the statistical analysis been performed appropriately and rigorously?

Reviewer #1: N/A

3. Have the authors made all data underlying the findings in their manuscript fully available?

Reviewer #1: Yes

4. Is the manuscript presented in an intelligible fashion and written in standard English?

Reviewer #1: Yes

Reviewer #1: This paper presents a study on a sample of charred bread dating back to the Bronze Age and found in Anatolia. The sample was analysed using optical and electron microscopy techniques, but also subjected to chemical analysis using EDX, FTIR, Raman and TG-DSC. The purpose of the analysis was to determine the raw materials used and the methods of preparation.

ABSTRACT: It seems clear to me, but I would add at least a mention of the type of analysis conducted, in order to make it more interesting for an audience with a more scientific background.

INTRODUCTION: The introduction clearly presents the importance of bread in human history, citing the main discoveries of ancient bread and reconstructing the history of this food with its highly symbolic and ritualistic power. The introduction also describes the site of origin of the sample analysed. In my opinion, given the type of journal chosen for publication, at least one paragraph dedicated to the scientific analysis of ancient bread is missing. This addition would serve to outline the state of the art in this field and justify the choice of analytical techniques used.

MATERIALS AND METHODS: In my opinion, all experimental details should be collected in this section and not in the discussion of results section. These details should also be completed, according to the instructions given in the specific comments.

RESULTS AND DISCUSSION: This section presents the results divided by technique. The idea is good, but the sections on SEM-EDX, FTIR, Raman and TG-DSC analyses need to be revised in terms of the interpretation of the results.

CONCLUSIONS: Perhaps a little long, some considerations could be brought forward to the ‘Results and discussion’ section.

ENGLISH LANGUAGE: The text is sufficiently clear, but needs some revisions to correct typos or a few incomplete sentences.

BIBLIOGRAPHY: There are only 2 self-citations out of 35 papers cited. As mentioned in the specific comments, I would suggest expanding the bibliography on the application of analytical techniques on similar samples of archaeological bread.

Specific comments:

1) Line 107

"Plant remains were obtained from the soil samples using a flotation system"

Please provide some more information about the flotation system used.

2) Lines 116-117

"The oval, flat bread measures 12 cm in diameter and 2.5 cm thick"

In my opinion, these details should be included in the materials and methods section.

3) Lines 127-128

"Conducted microscopic and elemental analyses to determine the carbonized bread sample's composition, production method, and plant content"

Please check this sentence, which appears to be incomplete.

4) Lines 128-132

Some instrumental details are missing, such as the energy used, the type of electrons used to collect the image, and the type of elemental analysis conducted (point? Area?). Was the sample metallised or observed as is? Were one or more fragments taken from the entire sample of bread found?

5) Lines 142-143

"Triticum dicoccum (emmer wheat) seed particles and bran were identified by the analysis (Fig. 7)."

What are the distinctive features that allowed the plant structures observed to be identified as emmer wheat?

6) Lines 159-160

"A detailed SEM analysis revealed starch granules of different morphologies and sizes (Fig. 10)."

Are the shape and size of these granules consistent with the identified plant species?

7) Lines 166-168

"The analysis revealed carbon (C) as the most dominant The element's concentration ranged from 49.4% to 59.8%, indicating that the organic structure was largely preserved through carbonization"

This sentence needs to be revised from a grammatical point of view. Furthermore, even if the organic structure had not been preserved during carbonisation, the result would still be to have a lot of carbon. In my opinion, the amount of carbon present cannot be used to assess the state of preservation of the bread sample.

8) Lines 169-170

"The high concentration of these two elements indicates that heat treatment and burial stabilize the carbon-based organic molecules"

The same considerations made in the previous point apply here. Even after exposure to heat, the initial carbon remains carbon...

9) Lines 176-177

"These multifaceted analyses demonstrate that the Küllüoba bread is not merely a carbonized residue but also preserves molecular-level traces of its plant components"

As already specified, EDX analysis cannot provide information at the molecular level, but only at the elemental level.

10) Line 183

"Vibrational spectroscopy is a nondestructive characterization technique"

That is not entirely correct. Some vibrational spectroscopy techniques are non-destructive, while others require, for example, grinding of the sample.

11) Line 184

"Fourier Transformed InfraRed (FTIR) spectra"

Transform, not Transformed

12) Lines 186-195

"Both spectra are complementary since FTIR detects vibrations based on electrical dipole moment changes, while Raman detects vibrations based on electrical polarizability changes. In other words, FTIR is sensitive to polar bonds affected by connected atoms or surrounding molecules. On the other hand, Raman is sensitive to the ability of the electron cloud to distort the bond (nonpolar covalent bonds, symmetric vibrational bonds, aromatic and conjugated systems, or heavy atomic bonds). During Raman measurement, the molecule's vibrational energy level briefly reaches a high-energy collision state and returns to a lower energy state by emitting a photon. The difference between the emitted photon and the excitant laser frequencies provides the Raman shift. The Raman shift corresponds to the frequency of the fundamental IR absorbance band of the bonds"

I am not sure it is necessary to introduce the theory of infrared and Raman spectroscopy in a paper of this kind. I assume that most readers know what we are talking about, or know where to find information if they do not already have it.

However, specific information on the analyses carried out is completely missing: brand and model of the instruments, analysis intervals, types of sources and detectors, resolution, number of scans, etc. For infrared spectroscopy: was an ATR technique used? All this information is fundamental in a scientific article, much more so than the theory of spectroscopy.

13) Line 202

"spectra of breads investigated in the modern literature"

Does the term ‘modern’ refer to literature or bread? From personal experience, I know that it is not easy to find studies that apply infrared spectroscopy to samples of ancient bread and that it is therefore common practice to compare the spectra obtained with those recorded on samples of modern bread. It also seems to me that the few studies of infrared spectroscopy on ancient bread have not been cited here.

14) Lines 205-206

"Deconvolution of the latter group of peaks shows that there are so many peaks that it is not possible to assign a meaning to specific molecules"

To be precise, individual peaks are not associated with molecules, but with specific vibrations within a molecule. Only by grouping together the information provided by multiple peaks is it possible to identify the molecule that produced them.

15) Lines 208-209

"those between 1750-2400 cm⁻¹ represent conjugated double or triple bonds between C, N, and/or O"

Looking at the spectrum, there do not appear to be any peaks in this region. The spectrum is extremely noisy, probably due to moisture in the sample or an unsatisfactory background correction. It would nevertheless be advisable to indicate at least the main peaks discussed in the text on the spectrum.

16) Lines 210-211

"but probably not all of them are related to the bread structure."

On what basis can this statement be justified?

17) Lines 222-224

"The lack of carbonyls may be natural, but the presence of a large number of additives (grains, leaves…) also suggests that carbonyl groups have probably disappeared in the structure."

This sentence is unclear.

18) Lines 242-244

"the attenuated C-H stretching modes suggest that the bread dough was prepared with a minimum amount of water and possibly without oil and/or salt."

However, line 236 stated that the attenuation of these peaks was due to the presence of oil and/or salt.

19) Line 264

"The thermogravimetric analysis result (Setaram, Labsys Evo analyzer)"

Here too, details such as heating rate and, above all, the type of atmosphere used (nitrogen, air, etc.) are missing.

20) Line 265

"shows a maximum mass loss of 13% when the temperature reaches 150°C."

Please explain this sentence more clearly.

21) Lines 266-267

"Therefore, it can be assumed that the Küllüoba bread is above 150°C. "

Please explain this sentence more clearly.

22) Lines 269-271

"which changes at endothermic values between 50 and 150°C, indicating that the chemical bonds in the bread are broken between these temperatures"

It is not necessarily the case that bonds are broken; even the evaporation of moisture generates an endothermic peak in DSC.

23) Lines 271-273

"The exothermic heat flow observed when the temperature rises above 160°C may be due to the formation of new bonds, suggesting that the possibility is likely baked at this temperature."

I honestly don't see the exothermic peak above 160°C. Could you indicate it clearly, perhaps with an arrow? On the contrary, I see a slight endothermic peak just before 250°C, linked to the corresponding weight loss.

24) Lines 315-317

"These samples, which show no significant differences compared to those inside the furnaces and are similar in content; they were probably scattered across the area as debris while cleaning the hearths"

Please check this sentence because it seems incomplete.

**Do you want your identity to be public for this peer review?** For information about this choice, including consent withdrawal, please see our For information about this choice, including consent withdrawal, please see our Privacy Policy .

Reviewer #1: No

---

## [Author Response · Author response to Decision Letter 1]

1 Jan 2026

Reviewer's Responses to Questions

Comments to the Author

1. Is the manuscript technically sound, and do the data support the conclusions?

Reviewer #1: Partly

Response: Following the reviewer’s suggestions, the manuscript has been revised and expanded, and the radiocarbon (C¹⁴) result for the bread has been included as a newly conducted analysis. In this context, we consider our study to be technically sound and to support its conclusions with important analytical results.

2. Has the statistical analysis been performed appropriately and rigorously?

Reviewer #1: N/A

3. Have the authors made all data underlying the findings in their manuscript fully available?

requires authors to make all data underlying the findings described in their manuscript fully available without restriction, with rare exception (please refer to the Data Availability Statement in the manuscript PDF file). The data should be provided as part of the manuscript or its supporting information, or deposited to a public repository. For example, in addition to summary statistics, the data points behind means, medians and variance measures should be available. If there are restrictions on publicly sharing data—e.g. participant privacy or use of data from a third party—those must be specified.

Reviewer #1: Yes

4. Is the manuscript presented in an intelligible fashion and written in standard English?

Reviewer #1: Yes

5. Review Comments to the Author

Reviewer #1: This paper presents a study on a sample of charred bread dating back to the Bronze Age and found in Anatolia. The sample was analysed using optical and electron microscopy techniques, but also subjected to chemical analysis using EDX, FTIR, Raman and TG-DSC. The purpose of the analysis was to determine the raw materials used and the methods of preparation.

ABSTRACT: It seems clear to me, but I would add at least a mention of the type of analysis conducted, in order to make it more interesting for an audience with a more scientific background.

Response to Abstract Comment: We appreciate the suggestion to make the abstract more appealing to a scientific audience. We have revised the abstract to explicitly list the specific analytical techniques employed in the study, including SEM-EDX, FTIR, Raman spectroscopy, and TGA-DSC. We believe this addition clearly highlights the archaeometric depth of the research.

INTRODUCTION: The introduction clearly presents the importance of bread in human history, citing the main discoveries of ancient bread and reconstructing the history of this food with its highly symbolic and ritualistic power. The introduction also describes the site of origin of the sample analysed. In my opinion, given the type of journal chosen for publication, at least one paragraph dedicated to the scientific analysis of ancient bread is missing. This addition would serve to outline the state of the art in this field and justify the choice of analytical techniques used.

Response to Introduction Comment: We appreciate the reviewer's valuable suggestion regarding the need for a scientific context. We agree that a dedicated paragraph outlining the 'state of the art' in ancient bread analysis was missing. We have inserted a new paragraph in the Introduction section that discusses the limitations of macroscopic examination and justifies the application of the specific archaeometric techniques used in this study (SEM, FTIR, Raman, TGA-DSC), citing relevant literature in the field.

MATERIALS AND METHODS: In my opinion, all experimental details should be collected in this section and not in the discussion of results section. These details should also be completed, according to the instructions given in the specific comments.

Response to Materials and Methods: We fully agree with the reviewer’s recommendation to consolidate all experimental details within the Materials and Methods section. In the revised manuscript, we have moved the descriptive details from the Results section to this section. Furthermore, we have significantly expanded the Materials and Methods section to include all missing technical specifications, instrument models, and operational parameters as requested in the specific comments.

RESULTS AND DISCUSSION: This section presents the results divided by technique. The idea is good, but the sections on SEM-EDX, FTIR, Raman and TG-DSC analyses need to be revised in terms of the interpretation of the results.

Response to Results and Discussion: We appreciate the reviewer’s positive feedback on the structure of this section and their constructive criticism regarding the data interpretation. We have extensively revised the Results and Discussion section to address the specific concerns raised in the detailed comments. Specifically, we have: Refined the interpretation of SEM-EDX results to focus on elemental composition rather than implying molecular stabilization or preservation based solely on carbon content. Re-evaluated the FTIR and Raman spectra descriptions to ensure accurate peak assignments and resolved contradictions regarding lipid/salt presence. Corrected the interpretation of thermal events in the TGA-DSC analysis, clarifying the distinction between moisture loss and chemical decomposition. These revisions ensure that our conclusions are scientifically robust and fully supported by the analytical data.

CONCLUSIONS: Perhaps a little long, some considerations could be brought forward to the ‘Results and discussion’ section.

Response to Conclusions: We have shortened this section by moving the detailed comparative discussions (such as comparisons with other archaeological sites like Çatalhöyük and Shubayqa 1) to the Results and Discussion section. The revised Conclusions section now focuses strictly on the primary findings and the broader implications of the study.

ENGLISH LANGUAGE: The text is sufficiently clear, but needs some revisions to correct typos or a few incomplete sentences.

Response to English language: We have carefully proofread the entire manuscript to identify and correct typos and complete any fragmented sentences. We believe the revised text is now grammatically accurate and flows more smoothly.

BIBLIOGRAPHY: There are only 2 self-citations out of 35 papers cited. As mentioned in the specific comments, I would suggest expanding the bibliography on the application of analytical techniques on similar samples of archaeological bread.

Response to Bibliography: We appreciate the reviewer’s constructive suggestion to broaden the bibliographic context. We have significantly expanded the reference list by including additional studies that apply similar analytical techniques (such as SEM, FTIR, and Raman) to archaeological bread samples. These new citations allow for a better integration of our findings with the current state of the art and provide a stronger basis for comparative discussion.

Specific note:

"We would like to inform you that the direct radiocarbon dating results for the bread sample, which were pending at the time of initial submission, have now been received. We have incorporated these new data into the revised manuscript, providing direct chronological evidence that further corroborates the stratigraphic dating."

1) Line 107

"Plant remains were obtained from the soil samples using a flotation system"

Please provide some more information about the flotation system used.

Response to Comment 1: We appreciate the reviewer’s request for more detailed information regarding the archaeobotanical recovery process. We have rewritten the relevant paragraph in the Materials and Methods section to provide a comprehensive description of the flotation system used. The revised text now clearly outlines the technical specifications of the flotation process.

2) Lines 116-117

"The oval, flat bread measures 12 cm in diameter and 2.5 cm thick"

In my opinion, these details should be included in the materials and methods section.

Response to Comment 2: We agree with the reviewer's suggestion that the physical dimensions of the sample are better suited for the Materials and Methods section. Accordingly, we have moved the sentence describing the bread's shape and measurements ('The oval, flat bread measures 12 cm in diameter and 2.5 cm thick') to that section.

3) Lines 127-128

"Conducted microscopic and elemental analyses to determine the carbonized bread sample's composition, production method, and plant content."

Please check this sentence, which appears to be incomplete.

Response to Comment 3: Thank you for bringing this grammatical oversight to our attention. We have revised the sentence to ensure it is complete and grammatically correct by converting it to the passive voice. The sentence now reads: 'Microscopic and elemental analyses were conducted to determine the carbonized bread sample's composition, production method, and plant content.

4) Lines 128-132

Some instrumental details are missing, such as the energy used, the type of electrons used to collect the image, and the type of elemental analysis conducted (point? Area?). Was the sample metallised or observed as is? Were one or more fragments taken from the entire sample of bread found?

Response to Comment 4: We have updated the Materials and Methods section to include all the missing instrumental details. The revised text now explicitly specifies the accelerating voltage (EHT), the type of electrons used for imaging (Secondary Electrons - SE), the nature of the EDX analysis (point and area measurements), and the sample preparation process (metallization). Additionally, we have clarified the sampling strategy, stating how the specific fragments were selected from the bread remains for analysis.

5) Lines 142-143

"Triticum dicoccum (emmer wheat) seed particles and bran were identified by the analysis (Fig. 7)."

What are the distinctive features that allowed the plant structures observed to be identified as emmer wheat?

Response to Comment 5: We used a combination of micromorphological features and the specific archaeobotanical context of the find to identify Triticum dicoccum, not just the SEM image.

In the SEM micrographs, we observed unique bran fragments (pericarp tissues) and coarse particle structures that exhibit morphological characteristics consistent with hulled wheat species. But as the reviewer points out, it can be hard to tell T. dicoccum apart from other Triticum species just by looking at carbonized flour fragments. Consequently, this diagnosis was robustly corroborated by the archaeobotanical analysis of the soil samples obtained from the immediate vicinity (hearths and floors) of the bread. Since Triticum dicoccum makes up most of the cereal assemblage in this area (as shown in the Archaeobotanical Results section) and the bran residues seen in SEM were all the same shape, the raw material was identified as emmer wheat.

We changed the sentence in the "Results and Discussion" section that talks about this to make it clear that the identification is based on this combined evidence, which makes it more scientifically accurate.

6) Lines 159-160

"A detailed SEM analysis revealed starch granules of different morphologies and sizes (Fig. 10)."

Are the shape and size of these granules consistent with the identified plant species?

Response to Comment 6: Yes, the morphology and size of the starch granules observed in the SEM analysis are consistent with the plant species identified in the sample, specifically Triticum dicoccum (emmer wheat) and Lens culinaris (lentil). We have clarified this consistency in the revised manuscript.

7) Lines 166-168

"The analysis revealed carbon (C) as the most dominant The element's concentration ranged from 49.4% to 59.8%, indicating that the organic structure was largely preserved through carbonization"

This sentence needs to be revised from a grammatical point of view. Furthermore, even if the organic structure had not been preserved during carbonisation, the result would still be to have a lot of carbon. In my opinion, the amount of carbon present cannot be used to assess the state of preservation of the bread sample.

Response to Comment 7: We agree with the reviewer’s assessment regarding both the grammatical structure and the interpretation of the results. We acknowledge that high carbon content is a natural outcome of the carbonization process and does not inherently prove the preservation of the structural integrity. Therefore, we have corrected the grammatical error and revised the sentence to state that the high carbon concentration reflects the carbonized nature of the organic material, rather than using it as an indicator of structural preservation.

8) Lines 169-170

"The high concentration of these two elements indicates that heat treatment and burial stabilize the carbon-based organic molecules"

The same considerations made in the previous point apply here. Even after exposure to heat, the initial carbon remains carbon...

Response to Comment 8: We accept the reviewer's valid point, which aligns with the previous comment. We acknowledge that the elemental presence of carbon and oxygen does not inherently prove the stabilization of specific molecular structures, as elemental carbon persists regardless of structural preservation. We have revised the sentence to remove the claim regarding 'stabilization' and instead stated that this composition is characteristic of carbonized organic matter derived from carbohydrates.

9) Lines 176-177

"These multifaceted analyses demonstrate that the Küllüoba bread is not merely a carbonized residue but also preserves molecular-level traces of its plant components"

As already specified, EDX analysis cannot provide information at the molecular level, but only at the elemental level.

Response to Comment 9: We fully agree with the reviewer’s correction. As noted, EDX analysis yields elemental data rather than molecular information. Since this sentence concludes the section on microscopic and elemental analyses (SEM-EDX), we have revised the text to accurately reflect the data obtained. We replaced the term 'molecular-level' with 'microstructural and elemental' to correctly describe the preservation traces detected by these specific methods.

10) Line 183

"Vibrational spectroscopy is a nondestructive characterization technique"

That is not entirely correct. Some vibrational spectroscopy techniques are non-destructive, while others require, for example, grinding of the sample.

Response to Comment 10: To ensure scientific accuracy, we have removed the term 'nondestructive' and redefined the technique based on its operational principle. The revised sentence now reads: Vibrational spectroscopy is a chemical characterization technique based on irradiation and time-limited excitation of the sample.

11) Line 184

"Fourier Transformed InfraRed (FTIR) spectra"

Transform, not Transformed

Response to Comment 11: We have corrected the phrase to 'Fourier Transform InfraRed' in accordance with the standard scientific nomenclature.

12) Lines 186-195

"Both spectra are complementary since FTIR detects vibrations based on electrical dipole moment changes, while Raman detects vibrations based on electrical polarizability changes. In other words, FTIR is sensitive to polar bonds affected by connected atoms or surrounding molecules. On the other hand, Raman is sensitive to the ability of the electron cloud to distort the bond (nonpolar covalent bonds, symmetric vibrational bonds, aromatic and conjuga

---

## [Decision Letter · Decision Letter 1]

24 Feb 2026

Archaeometric analysis of Early Bronze Age bread from Küllüoba Höyük

PONE-D-25-50027R1

Dear Dr. kavak,

We’re pleased to inform you that your manuscript has been judged scientifically suitable for publication and will be formally accepted for publication once it meets all outstanding technical requirements.

Kind regards,

Enrico Greco

Academic Editor

PLOS One

Additional Editor Comments (optional):

Reviewers' comments:

Reviewer's Responses to Questions

**Comments to the Author**

Reviewer #1: All comments have been addressed

2. Is the manuscript technically sound, and do the data support the conclusions?

Reviewer #1: Yes

3. Has the statistical analysis been performed appropriately and rigorously?

Reviewer #1: N/A

4. Have the authors made all data underlying the findings in their manuscript fully available?

Reviewer #1: Yes

5. Is the manuscript presented in an intelligible fashion and written in standard English?

Reviewer #1: Yes

Reviewer #1: I have carefully re-read the paper and it has improved substantially, both in terms of content and in the structure and readability of the text. All the issues I previously raised appear to have been fully addressed.

I have no further comments and believe the manuscript is now suitable for publication.

**Do you want your identity to be public for this peer review?** For information about this choice, including consent withdrawal, please see our For information about this choice, including consent withdrawal, please see our Privacy Policy .

Reviewer #1: No

---

## [Editor Report · Acceptance letter]

PONE-D-25-50027R1

PLOS One

Dear Dr. kavak,

I'm pleased to inform you that your manuscript has been deemed suitable for publication in PLOS One. Congratulations! Your manuscript is now being handed over to our production team.

Kind regards,

on behalf of

Dr. Enrico Greco

Academic Editor

PLOS One